# Faster Discovery of Neural Architectures by Searching for Paths in a Large Model

## Abstract

We propose *Efficient Neural Architecture Search* (ENAS), a faster and less expensive approach to automated model design than previous methods. In ENAS, a controller learns to discover neural network architectures by searching for an optimal path within a larger model. The controller is trained with policy gradient to select a path that maximizes the expected reward on the validation set. Meanwhile the model corresponding to the selected path is trained to minimize the cross entropy loss. On the Penn Treebank dataset, ENAS can discover a novel architecture thats achieves a test perplexity of 57.8, which is state-of-the-art among automatic model design methods on Penn Treebank. On the CIFAR-10 dataset, ENAS can design novel architectures that achieve a test error of 2.89%, close to the 2.65% achieved by standard NAS (Zoph et al., 2017). Most importantly, our experiments show that ENAS is more than 10x faster and 100x less resource-demanding than NAS.

## 1 Introduction

Neural architecture search (NAS) has been applied successfully to design model architectures for image classification and language modeling (Zoph & Le, 2017; Baker et al., 2017a; Bello et al., 2017b; Zoph et al., 2017; Cai et al., 2017). NAS however is computationally expensive and time consuming: for example, Zoph et al. (2017) use 450 GPUs and train for 3-4 days. Meanwhile, using less resources tends to produce less compelling results (Negrinho & Gordon, 2017; Baker et al., 2017a).

The main computational bottleneck of NAS is the training of each child model to convergence to measure its accuracy. We believe that it is very inefficient and wasteful to train every child model and then throw away all the trained weights even though the child models have much in common.

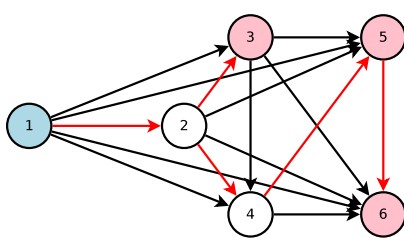

Figure 1: The graph represents the entire search space while the red arrows define a model in the search space, which is decided by a controller. Here we assume that node 1 is the input to the model whereas nodes 3, 5, and 6 are the outputs of the model.

The goal of this work is to remove this inefficiency by enabling more sharing between the child models. This idea is similar to the concept of weight inheritance in neuro-evolution (e.g., Real et al. (2017)). To understand our method, we first need to understand the standard NAS. In standard NAS (Zoph & Le, 2017; Baker et al., 2017a), an RNN controller is trained by policy gradient to search for a good architecture, which is basically a computational graph. Our observation is that all of the graphs, that NAS has iterated over, can be viewed as sub-graphs of a larger graph. In other words, we can represent the space of these graphs as a *single* directed acyclic graph (DAG). As illustrated in Figure 1, a neural network architecture can be found by taking a subset of edges in this DAG. This design is advantageous because it enables sharing parameters among all architectures in the search space.

Our method requires the training of a path within a large model together with the controller. The controller is trained with policy gradient to select a path that maximizes the expected reward on the validation set. Meanwhile the model corresponding to the selected path is trained to minimize a cross-entropy loss. In view of its efficiency, we name our method *Efficient Neural Architecture Search* (ENAS).

ENAS is both theoretically sound and empirically efficient. Using Lyapunov functions (Bottou, 1991), we can show that the training of ENAS converges almost surely. Empirically, on CIFAR-10, ENAS achieves the error rate of 2.89%, compared to 2.65% by NAS. On Penn Treebank, ENAS discovers an architecture that achieves the perplexity of 57.8, which outperforms NAS's perplexity of 62.4 (Zoph & Le, 2017). In all of our experiments, ENAS takes less than 16 hours to train, whilst running on a single Nvidia GTX 1080Ti GPU. Compared to NAS, our method achieves a 10X reduction of search time and 100X reduction of computational resources.

## 2  RELATED WORK

There is growing interest in improving the efficiency of neural architecture search. Concurrent to our work are the promising ideas of using learning curve prediction to skip bad models (Baker et al., 2017b), predicting the accuracies of models before training (Deng et al., 2017), using iterative search method for architectures of growing complexity (Liu et al., 2017a; Elsken et al., 2017), or using hierarchical representation of architectures (Liu et al., 2017b). Our method is also inspired by the concept of weight inheritance in neuro-evolution, which has been demonstrated to have positive effects at scale (Real et al., 2017).

Closely related to our method are other recent approaches that avoid training each architecture from scratch, such as convolutional neural fabrics – ConvFabrics (Saxena & Verbeek, 2016) and SMASH (Brock et al., 2017). These methods are more computationally efficient than standard NAS. However, the search space of ConvFabrics is not flexible enough to include novel architectures, e.g. architectures with arbitrary skip connections as in Zoph & Le (2017). Meanwhile, SMASH can design interesting architectures but requires a hypernetwork (Ha et al., 2017) to generate the weights, conditional on the architectures. While a hypernetwork can efficiently *rank* different architectures, as shown in the paper, the real performance of each network is different from its performance with parameters generated by a hypernetwork. Such discrepancy in SMASH can cause misleading signals for reinforcement learning. Even more closely related to our method is PathNet (Fernando et al., 2017), which uses evolution to search for a path inside a large model for transfer learning between Atari games.

## 3  EFFICIENT NEURAL ARCHITECTURE SEARCH

In the following, we will first present our search space to design recurrent cells for recurrent networks. We will then explain how to train, and infer with the controller. Finally, we will explain our search space to design convolutional architectures.

### 3.1  DESIGNING RECURRENT CELLS

To facilitate our discussion of ENAS, we first describe how we employ ENAS to design a recurrent cell. The search space of ENAS is a Directed Acyclic Graph as mentioned in Section 1. The DAG has $N$ nodes where the edges represent the flow of information between these $N$ nodes. Similar to NAS (Zoph & Le, 2017), ENAS has a controller RNN, which decides which edges are activated and which computations are performed at each node.

To create a recurrent cell, the controller RNN samples $N$ blocks of decisions. Here we illustrate the ENAS mechanism via a simple example for a recurrent cell with $N = 4$ computational nodes. Let $\mathbf{x}_t$ be the input signal for a recurrent cell (e.g., word embedding), and $\mathbf{h}_{t-1}$ be the output from the previous time step. The example cell, which we visualize in Figure 2, is sampled as follows.

1. At node 1: The controller first samples an activation function. In our example in Figure 2, it chooses the tanh activation function, which means that node 1 of the recurrent cell should compute $h_1 = \tanh\left(\mathbf{x}_t \cdot \mathbf{W}^{(\mathbf{x})} + \mathbf{h}_{t-1} \cdot \mathbf{W}_1^{(\mathbf{h})}\right)$.

2. At node 2: The controller then samples a previous index and an activation function. In our example, it chooses the previous index 1 and the activation function ReLU. Thus, node 2 of the recurrent cell should compute $h_2 = \text{ReLU}(h_1 \cdot \mathbf{W}_{2,1}^{(\mathbf{h})})$.

3. At node 3: The controller again samples a previous index and an activation function. In our example, it chooses the previous index 2 and the activation function ReLU. Therefore, $h_3 = \text{ReLU}(h_2 \cdot \mathbf{W}_{3,2}^{(\mathbf{h})})$.

4. At node 4: The controller again samples a previous index and an activation function. In our example, it chooses the previous index 1 and the activation function tanh, leading to $h_4 = \tanh\left(h_1 \cdot \mathbf{W}_{4,1}^{(\mathbf{h})}\right)$.

5. For the output, we simply average all the loose ends, which are the nodes that are not input to any other nodes. In our example, since the indices 3 and 4 were never sampled to be the input for any node, the recurrent cell uses their average $(h_3 + h_4)/2$ as its output.

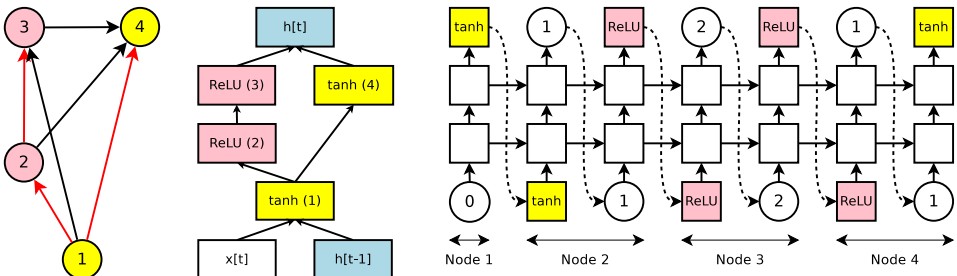

Figure 2: An example of a recurrent cell in our search space with 4 computational nodes. *Left:* The computational DAG that corresponds to the recurrent cell. The red edges represent the activated path. *Middle:* The recurrent cell. *Right:* The outputs of the controller RNN that result in the cell in the middle and the DAG on the left. Note that nodes 3 and 4 are never sampled by the RNN, so their results are averaged and are treated as the cell's output.

In the example above, we note that for each pair of nodes $j < \ell$, there is an independent parameter matrix $\mathbf{W}_{\ell,j}^{(\mathbf{h})}$. As shown in the example, the controller decides which parameter matrices are used, by choosing the previous indices. Therefore, in ENAS, all recurrent cells in a search space share the same set of parameters.

Our search space includes an exponential number of configurations. Specifically, if the recurrent cell has $N$ nodes and we allow 4 activation functions (namely tanh, ReLU, identity, and sigmoid), then the search space has $4^N \times N!$ configurations. In our experiments, $N = 12$, which means there are approximately $10^{15}$ models in our search space.

## 3.2 TRAINING AND INFERENCE WITH ENAS

Our controller network is a two-layer LSTM (Hochreiter & Schmidhuber, 1997) which samples decisions via softmax classifiers. The controller network samples these decisions in an autoregressive fashion: the decision in the previous step is fed as input embedding into the next step. At the first step, the controller network receives an empty embedding as input.

In ENAS, there are two sets of learnable parameters: the parameters of the controller LSTM, denoted by $\theta$, and the shared parameters of the child models, denoted by $\omega$. The training procedure of ENAS consists of two alternating phases. The first phase trains $\omega$, the shared parameters of the child models, on a whole pass through the training data set, and the second phase trains $\theta$, the parameters of the controller LSTM, for a fixed number of steps. These two phases are alternated during the training of ENAS. In our experiments with the Penn Treebank dataset, for each phase of training $\omega$, we train $\omega$ for 450 steps with SGD, each

on a batch of 64 examples, where gradients are computed using back-propagation through time, truncated to 35 time steps. Meanwhile, for each phase of training $\theta$, we train it for 2000 steps with the Adam optimizer and REINFORCE. More details of their training are as follows.

**Training the shared parameters $\omega$ of the child models.** In this step, we fix the policy $\pi(\mathbf{m}; \theta)$ and perform stochastic gradient descent (SGD) updates on $\omega$ to minimize the expected loss function $\mathbb{E}_{\mathbf{m} \sim \pi(\mathbf{m}; \theta)} [\mathcal{L}(\mathbf{m}; \omega)]$. Here, $\mathcal{L}(\mathbf{m}; \omega)$ is the standard cross-entropy loss, computed on a minibatch of training data, with a model $\mathbf{m}$ sampled from $\pi(\mathbf{m}; \theta)$. The gradient is computed via the Monte Carlo estimate

$$\nabla_\omega \mathbb{E}_{\mathbf{m} \sim \pi(\mathbf{m}; \theta)} [\mathcal{L}(\mathbf{m}; \omega)] \approx \frac{1}{M} \sum_{i=1}^{M} \nabla_\omega \mathcal{L}(\mathbf{m}_i, \omega). \tag{1}$$

Note that Eqn 1 provides an unbiased estimate of the gradient $\nabla_\omega \mathbb{E}_{\mathbf{m} \sim \pi(\mathbf{m}; \theta)} [\mathcal{L}(\mathbf{m}; \omega)]$. Therefore, according to Chapter 3.3.2 of Bottou (1991), with an appropriate learning schedule, the SGD updates of $\omega$ converge almost surely. While convergence is guaranteed, these updates on $\omega$ have an inherently larger variance than SGD performed on a fixed model $\mathbf{m}$. Nevertheless, we find that $M = 1$ works just fine, i.e. we can update $\omega$ using the gradient from *any single model* $\mathbf{m}$ sampled from $\pi(\mathbf{m}; \theta)$. As mentioned, we train $\omega$ for a whole pass through the training data set.

**Training the parameters of $\theta$ of the controller LSTM.** In this step, we fix $\omega$ and update the policy parameters $\theta$, aiming to maximize the expected reward $\mathbb{E}_{\mathbf{m} \sim \pi(\mathbf{m}; \theta)} [R(\mathbf{m}, \omega)]$. We employ the Adam optimizer (Kingma & Ba, 2015), for which the gradient is computed using REINFORCE (Williams, 1992), with a moving average baseline to reduce variance.

We compute the reward $R(\mathbf{m}, \omega)$ on *the validation set*, rather than the training set to encourage ENAS, to select the model that generalizes well rather than the model that overfits the training set well. In our experiments with language modeling, the reward function is $c/\text{valid\_ppl}$, where the perplexity is computed on a minibatch, also sampled from the validation set. Later, in our experiments with image classification, the reward function is the classification accuracy on a minibatch of images sampled from the validation set.

**Inference.** We discuss how to derive novel architectures from a trained ENAS model. Following Zoph & Le (2017), we sample several models from the trained policy $\pi(\mathbf{m}, \theta)$. For each sampled model, we compute its reward on a single minibatch sampled from the validation set. We then take the model with the highest reward to re-train from scratch.

## 3.3 Designing Convolutional Models

We now discuss the search space for convolutional architectures. Recall that in the search space of the recurrent cell, the controller RNN samples two decisions at each decision block: 1) what previous node to connect to and 2) what activation function to use. In the search space for convolutional models, the controller RNN also samples two sets of decisions at each decision block: 1) what previous node to connect to and 2) what computation operation to use. The decision of what previous node to connect to allows the model to form skip connections (He et al., 2016; Zoph & Le, 2017); whereas the decision of what computation operation to use sets a particular layer into convolution or average pooling or max pooing. These decisions help construct a layer in the convolutional model.

To be even more flexible, instead of deciding if a particular layer is convolution or average pooling or max pooling, we change the search space to blend all these choices together. To achieve this, we treat all operations such as convolution, average pooling, and max pooling as channels, and allow the controller to select a mask over these channels. For example, in our experiments, we allow the controller to choose a mask over how many conv1x1, conv3x3, conv5x5, conv7x7, average pooling, max pooling operations to use (see Figure 3). Each operation at each layer in our network has its own convolutional parameters. During each data pass, only the parameters corresponding to the active channels are used and updated.

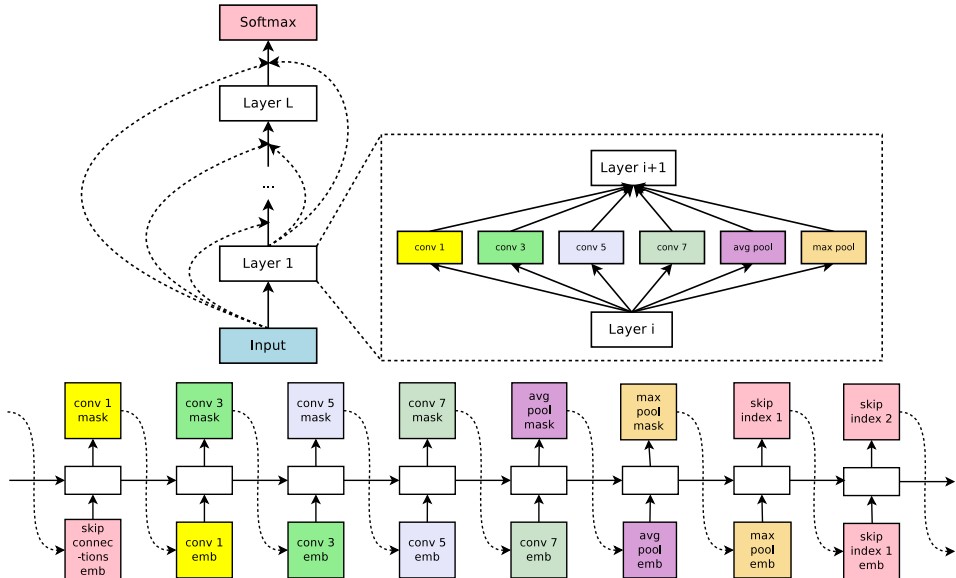

Figure 3: Our parameter sharing scheme between convolutional models. *Top:* The network of $N$ layers, where each layer has 6 channels as described. *Bottom:* A block of the controller network, which consists of 6 binary masks, followed by the steps that sample skip connections.

For the skip connections, at layer $k$, up to $k-1$ mutually distinct previous indices are sampled. For example, at layer $k = 5$, let's suppose the controller samples $\{2, 4\}$, then the outputs of layer 2 and layer 4 will be concatenated and sent to layer 5 via skip connections.

In our experiments with the CIFAR-10 dataset, the training of the controller LSTM and the child models are also alternating. For each phase of training $\theta$, we train it for 2000 steps with the Adam optimizer and REINFORCE. Meanwhile, for each phase of training the shared parameter $\omega$, we train it for 450 minibatches, each has 100 images, using Nesterov Momentum (Nesterov, 1983).

## 3.4 Designing Convolutional Cells

An alternative to designing the entire convolutional network is to design smaller modules and then fit repeats of them together (Zoph et al., 2017; Zhong et al., 2017). Figure 4 illustrates this approach, where the convolutional cell and reduction cell architectures are to be designed. We now discuss how to use ENAS to search for the architecture of these cells.

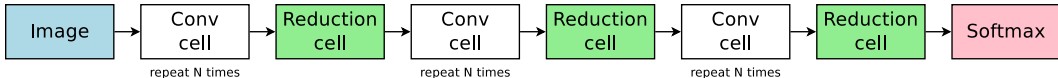

Figure 4: The organization of the convolution cell and reduction cell to form a network.

Following Zoph et al. (2017), we sample both our convolutional cell and our reduction cell using an RNN controller. Specifically, at each decision block, the controller RNN samples for two sets of decisions: 1) two previous nodes to be used as inputs to the current block and 2) two operations to respectively apply to the two sampled nodes. We allow the following 5 operations: identity, separable convolution with kernel size 3x3 and 5x5, and average pooling and max pooling with kernel size 3x3. Note that each cell receives two input nodes, indexed by node 1 and node 2, corresponding to the outputs of the two previous cells in the entire network. Figure 5 depicts an example run of our controller in the search space with 4 nodes. As with our other search spaces, each operation in each cell has its own parameters. During

each data pass, only the relevant parameters are used and trained with their corresponding gradients.

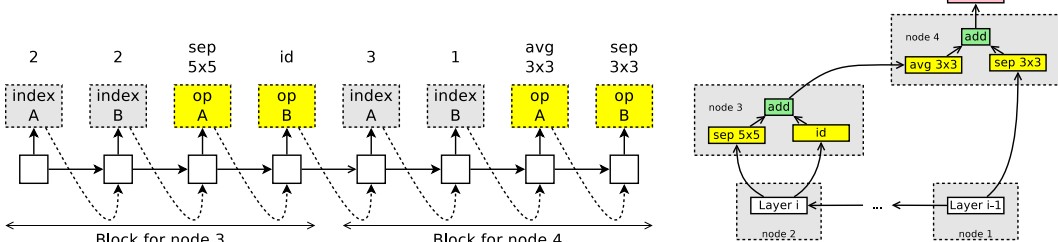

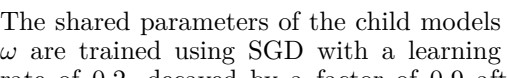

Figure 5: An example run of the controller for our convolutional cell search space. *Left:* the controller's outputs. Note that in our search space for convolutional cells, node 1 and node 2 respectively correspond to the output of the two previous layers, so the controller does not have to make any decisions for them. *Right:* the convolutional cell according to the controller's sample.

## 4 EXPERIMENTS

In the following, we will show our experimental results with ENAS to design recurrent cells on the Penn Treebank dataset and to design convolutional architectures on the CIFAR-10 dataset. We then present an ablation study which shows the role of ENAS in discovering novel architectures, as well as details regarding the efficiency of ENAS.

### 4.1 LANGUAGE MODEL WITH PENN TREEBANK

**Dataset.** We first apply ENAS to the task of language modeling, whose goal is predict next words in a text given a history of previous words, by fitting a probabilistic model over sentences. We use the Penn Treebank dataset (PTB) (Marcus et al., 1994), a well-studied language modeling benchmark. In this experiment, we show that ENAS discovers a novel recurrent cell which, without extensive hyper-parameters tuning, can outperform models with the same or more parameters.

**Training details.** Our controller is trained using Adam (Kingma & Ba, 2015), with a learning rate of 0.001. We use a tanh constant of 2.5 and a temperature of 5.0 for the sampling logits (Bello et al., 2017a). We also add the controller's sample entropy to the reward, with a weight of $10^{-4}$. Additionally, we augment the simple transformations between the constructed recurrent cell's nodes with highway connections (Zilly et al., 2017). For instance, instead of having $h_2 = \text{ReLU}(h_1 \cdot \mathbf{W}_{2,1}^{(\mathbf{h})})$ as shown in the example from Section 3.1, we have $h_2 = c_2 \otimes \text{ReLU}(h_1 \cdot \mathbf{W}_{2,1}^{(\mathbf{h})}) + (1 - c_2) \otimes h_1$, where $c_2 = \text{sigmoid}(h_1 \cdot \mathbf{W}_{2,1}^{(\mathbf{c})})$. More details can be found in Appendix A. A novel RNN cell found in our search space is shown in Figure 6.

The shared parameters of the child models $\omega$ are trained using SGD with a learning rate of 0.2, decayed by a factor of 0.9 after every 3 epochs starting at epoch 15, for a

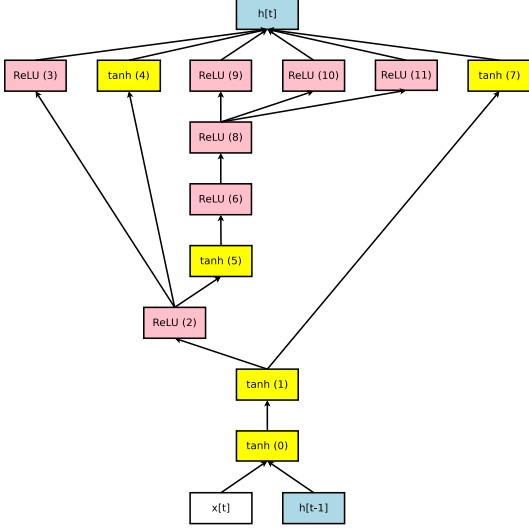

Figure 6: The best RNN cell discovered by ENAS on the Penn Treebank dataset.

total of 150 epochs. During the architecture search process, following Melis et al. (2017), we randomly reset the starting state with probability of 0.001. We also tie the model's word embeddings matrix with its softmax matrix (Inan et al., 2017). When retraining the architecture recommended by the controller, however, we use variational dropout (Gal & Ghahramani, 2016), an $\ell_2$ regularization with weight decay of $10^{-7}$, and a state slowness regularization of 0.0005 (Merity et al., 2017).

**Results.**  Running on a single Nvidia GTX 1080Ti GPU, ENAS finds the recurrent cell in less than 10 hours. This cell is depicted in Figure 6. Table 1 presents our results in comparison with other methods. The ENAS cell, with 24M parameters, outperforms the NAS cell and has a similar performance to the LSTM model that uses extensive hyper-parameters tuning (Melis et al., 2017), which we did not do.

Our ENAS cell has a few interesting properties. First, all non-linearities in the cell are either ReLU or tanh, even though the search space also has two other functions: identity and sigmoid. We suspect this cell is a local optimum, similar to the observations by Zoph & Le (2017). When we randomly pick some nodes and switch the non-linearity into identity or sigmoid, the perplexity increases up to 8 points. When we randomly, swith some ReLU nodes into tanh or vice versa, the perplexity also increases, but only up to 3 points.

| Method | Parameters (million) | Test Perplexity |
|---|---|---|
| LSTM+Vanilla Dropout (Zaremba et al., 2014) | 66 | 78.4 |
| LSTM+VD+MC (Gal & Ghahramani, 2016) | 66 | 73.4 |
| LSTM+WT (Inan et al., 2017) | 51 | 68.5 |
| Recurrent Highway Network (Zilly et al., 2017) | 24 | 66.0 |
| LSTM+Hyper-parameters Search (Melis et al., 2017) | 24 | 59.5 |
| LSTM+AWD (Merity et al., 2017) | 24 | 52.8 |
| LSTM+AWD+Dynamic Eval (Krause et al., 2017) | **24** | **51.1** |
| NAS (Zoph & Le, 2017) | 32 | 67.9 |
| NAS+VD | 25 | 64.0 |
| NAS+VD+WT (Zoph & Le, 2017) | 54 | 62.4 |
| ENAS cell | **24** | **57.8** |

Table 1: Test perplexity on Penn Treebank of ENAS and other approaches.  VD = Variational Dropout; WT = Weight Tying; MC = Monte Carlo sampling.

## 4.2 Image Classification on CIFAR-10

**Dataset.**  The CIFAR-10 dataset (Krizhevsky, 2009) consists of 50,000 training images and 10,000 test images. We use the standard data pre-processing and augmentation techniques, i.e., subtracting the mean and dividing the standard deviation from each channel computed on the training images, centrally padding the training images to $40 \times 40$ and randomly cropping them back to $32 \times 32$, and randomly flipping them horizontally.

**Search spaces.**  In Section 3.3, we present a search space for convolutional architectures in which the controller can make decisions over skip connections and the mask over the channels. To improve our results, we additionally explore two restricted versions of this search space: one where the controller only needs to make decisions of the mask over the channels and one where the controller only needs to make decisions of the skip connections. More details are available in Appendix B.

1. **Searching for the masks over channels.** We fix a pattern of skip connections and search for the masks at each branch and each layer in a 12-layer network. The pattern that we use is the dense pattern (Huang et al., 2016).
2. **Searching for skip connections.** We force all the convolutions to have the filter size of $3 \times 3$, and only search for the skip connections.

3. **Searching for convolutional and reduction cells.** We search for both cells as discussed in Section 3.4.

**Training details.** The shared parameters $\omega$ are trained with Nesterov momentum (Nesterov, 1983), where the learning rate follows the cosine schedule with $l_{\max} = 0.05$, $l_{\min} = 0.001$, $T_0 = 10$ and $T_{\mathrm{mul}} = 2$ (Loshchilov & Hutter, 2017). Each architecture search is run for $10 + 20 + 40 + 80 + 160 = 310$ epochs. The parameter $\omega$ is initialized from a scaled Gaussian as described in He et al. (2015). We also apply an $\ell_2$ weight decay of $10^{-4}$. The same settings are employed to train the architectures recommended by the controller.

The policy parameters $\theta$ are initialized uniformly in $[-0.1, 0.1]$, and trained with the Adam optimizer at a learning rate of $10^{-3}$. We additionally utilize three techniques to prevent the premature convergence of REINFORCE. First, we apply a temperature $\tau = 5.0$ and a tanh constant $c = 2.5$ to the controller's logits. Second, we add to the reward the entropy term of the controller's samples weighted by $\lambda_{\mathrm{ent}} = 0.1$, which discourages convergence (Williams & Peng, 1991). Lastly, we enforce the sparsity in the skip connections by adding to the reward the Kullback-Leibler divergence between: 1) the skip connection probability between any two layers and 2) our chosen probability $\rho = 0.4$, which represents the prior belief of a skip connection being formed. The KL divergence term is weighted by $\lambda_{\mathrm{kl}} = 0.5$.

**Tricks to stabilize and improve training.** We find the following tricks crucial for achieving good performance with ENAS.

- **Structure of Convolutional Layers.** Each convolutional operation in our method is followed by a batch normalization (Ioffe & Szegedy, 2015) and then a ReLU layer. We find the alternate setting of batch norm-conv-ReLU (Zoph et al., 2017) to have worse results.

- **Stabilizing Stochastic Skip Connections.** If a layer receives skip connections from multiple layers before it, then these layers' outputs are concatenated in their depth dimension, and then a convolution of filter size $1 \times 1$ (followed by a batch normalization layer and a ReLU layer) is performed to ensure the number of output channels is still equal to $C$.

- **Global Average Pooling.** After the final convolutional layer, we average all the activations of each channel and then pass them to the Softmax layer. This trick was introduced by Lin et al. (2013), with the purpose of reducing the number of parameters in the dense connection to the Softmax layer to avoid overfitting.

The last two tricks are extremely important, since the gradient updates of the shared parameters $\omega$, as described in Eqn 1, have a very high variance. In fact, we find that without these last two tricks, the training of ENAS is very unstable.

**Results.** Table 2 summarizes the test errors of ENAS and other approaches. As can be seen from the table, ENAS successfully found several architectures that outperform other automatic model designing approaches with the same usage of computing resource. In particular, in the general search space, ENAS takes 15.6 hours to find a model that achieves 4.23% error rate on CIFAR-10. This model outperforms all but one model reported by Zoph & Le (2017), while taking 30x less time and using 800x less computing resource to discover.

In the restricted search space over the masks, ENAS takes 11.6 hours to find a model that achieves the test error of 4.35%. The resulting model, depicted in Figure 7-*Top Left*, almost always has 64 or 96 channels at each branch and each layer, indicating that the controller does *not* choose to activate all blocks. This is the desired behavior, as activating all channels would over-parametrize the model and result in overfitting. Moreover, the fact that the model found in a restricted search space has similar performance to the model found in the general search space indicates that ENAS can discover skip connection patterns that are comparable to the dense pattern, which is the state-of-the-art human-designed architecture on CIFAR-10 (Huang et al., 2016).

In the restricted search space over skip connections, ENAS takes 12.4 hours to discover the pattern of skip connections depicted in Figure 7-*Top Right*. This pattern has the property

| Method | Depth | Parameters (million) | Error (%) |
|---|---|---|---|
| DenseNet-BC (Huang et al., 2016) | 190 | 25.6 | 3.46 |
| DenseNet + Shake-Shake 26 2x96d (Gastaldi, 2016) | 26 | 26.2 | 2.86 |
| DenseNet + CutOut (DeVries & Taylor, 2017) | 26 | 26.2 | **2.56** |
| Budgeted Super Nets (Veniat & Denoyer, 2017) | 16 | − | 9.21 |
| ConvFabrics Dense (Saxena & Verbeek, 2016) | 16 | 21.2 | 7.43 |
| Macro NAS with Q-Learning (Baker et al., 2017a) | 11 | 11.2 | 6.92 |
| Net Transformation (Cai et al., 2017) | 17 | 19.7 | 5.70 |
| FractalNet (Larsson et al., 2017) | 21 | 38.6 | 4.60 |
| SMASH (Brock et al., 2017) | 211 | 16.0 | 4.03 |
| NAS (Zoph & Le, 2017) | 39 | 7.1 | 4.47 |
| NAS + more filters (Zoph & Le, 2017) | 39 | 37.4 | **3.65** |
| ENAS, general search space | 18 | 34.9 | 4.23 |
| ENAS, search for masks | 12 | 12.6 | 4.35 |
| ENAS, search for skip connections | 15 | 14.1 | 5.04 |
| ENAS, search for skip connections + 512 channels | 15 | 38.0 | **3.87** |
| MicroNAS with Q-Learning (Zhong et al., 2017) | 24 | − | 3.60 |
| NASNet-A (Zoph et al., 2017) | 20 | 3.3 | 3.41 |
| NASNet-A (Zoph et al., 2017) + CutOut | 20 | 3.3 | **2.65** |
| ENAS, search for cells | 20 | 4.6 | 3.54 |
| ENAS, search for cells + CutOut | 20 | 4.6 | **2.89** |

Table 2: Classification error rates of ENAS and other methods on CIFAR-10. In this table, the first block presents the state-of-the-art models, all of which are designed by human experts. The second block presents various approaches that design the entire network. ENAS outperforms all these methods but NAS, which requires much more computing resource and time. The last block presents techniques that design modular cells which are used to build a large model. ENAS outperforms MicroNAS, which uses 32 GPUs to search, and achieves similar performance with NASNet-A.

that skip connections are formed much more densely at higher layers than at lower layers, where most connections are only between consecutive layers. The model has the test error of 5.04%, which is slightly worse than the one found in the restricted search space over masks. However, if we increase the number of output channels from 256 to 512, then the network achieves the test error of 3.87%.

In the search space over cells, ENAS takes 11.5 hours to discover the convolution cell and the reduction cell, which are shown in Figure 7-*Bottom*. With the convolutional cell replicated for 6 times, ENAS achieves 3.54% test error, which is on par with the 3.41% error of NASNet-A (Zoph et al., 2017). With CutOut (DeVries & Taylor, 2017), ENAS's error decreases to 2.89%, compared to 2.65% by NASNet-A. However, as discussed in Appendix B, our ENAS cells are trained for only 310 epochs, compared to 600 epochs by NAS.

### 4.3 The Importance and Efficiency of ENAS

**Sanity Check and Ablation Study.** To understand the role of ENAS, we carry out two control experiments. In the first study, we uniformly at random pick a configuration of channels and skip connections and just train a model. As a result, about half of the channels and skip connections are selected, resulting in a model with 47.1M parameters and an error rate of 5.86%. This error rate is significantly worse than the models designed by ENASand has many more parameters. In the second study, we only train $\omega$ and do not update the controller. The effect is similar to dropout with a rate of 0.5 on both the channels and the skip connections. At convergence, the model has the error rate of 11.92%. On the validation set, the ensemble of 250 Monte Carlo configurations of the trained model could only reach 8.99% test error rate. We therefore conclude that the appropriate training of the ENAS controller is crucial for good performance.

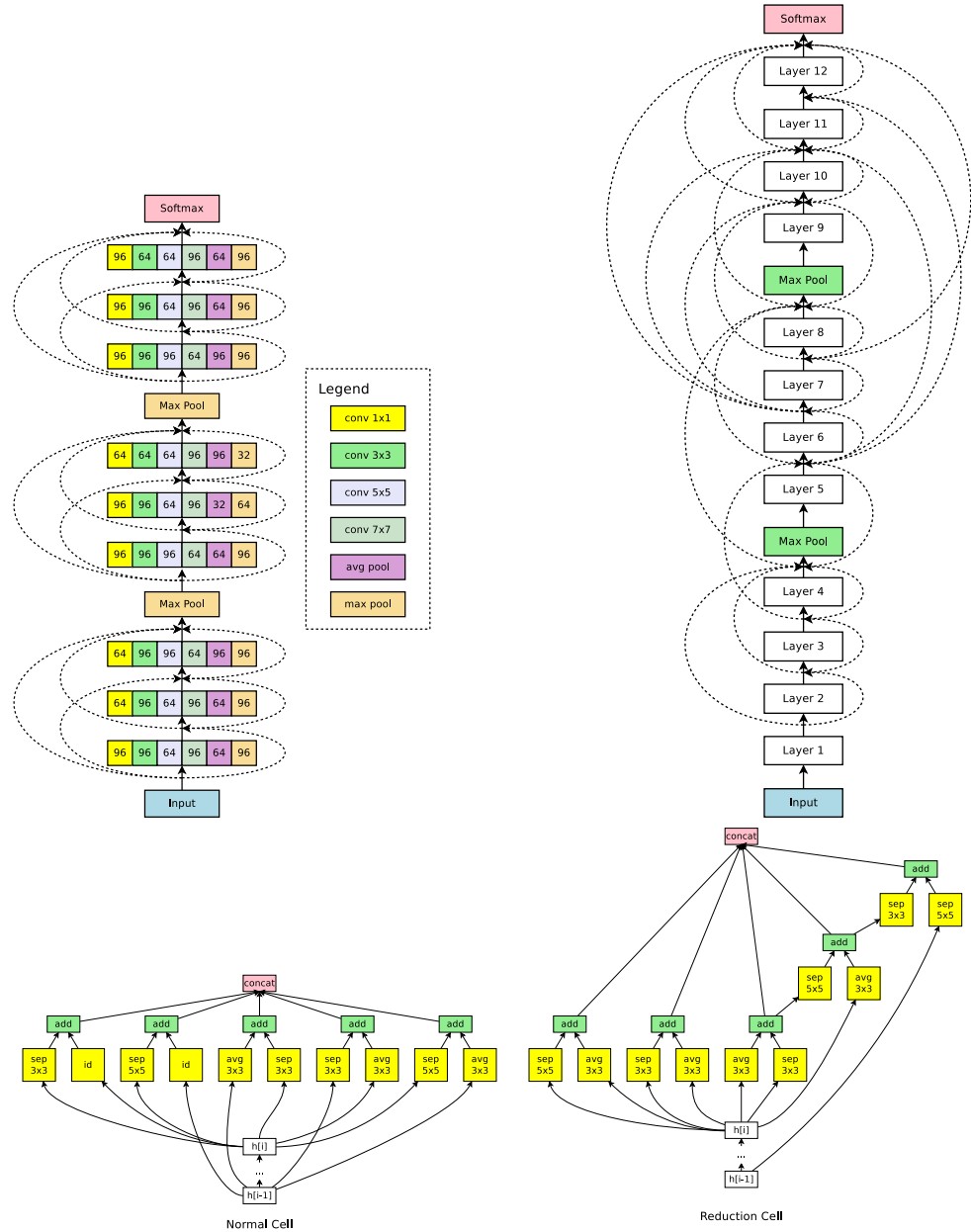

Figure 7: *Top left:* An architecture found by ENAS for CIFAR-10 for the search space over channels. *Top right:* An architecture found by ENAS for CIFAR-10 in the search space over skip connections. *Bottom:* the normal cell and the reduction cell found in the search space over cells.

**Efficiency of ENAS.** Table 3 further compares of the time and resources needed for ENAS to discover good architectures on CIFAR-10 with other methods. In all of our experiments, ENAS takes less than 15 hours to train, whilst running on a single Nvidia GTX 1080Ti GPU. Compared to NAS by Zoph & Le (2017); Zoph et al. (2017), our method achieves a 10X reduction of search time and 100X reduction of computational resources.

---

[1]We took the authors' published code, ran it for a few epochs and interpolated the running time.

| Method | GPUs | Time (days) |
|---|---|---|
| Macro NAS with Q-Learning (Baker et al., 2017a) | 10 | 8-10 |
| Net Transformation (Cai et al., 2017) | 5 | 3 |
| SMASH (Brock et al., 2017) | 1 | 1.5[1] |
| Micro NAS with Q-Learning (Zhong et al., 2017) | 32 | 3 |
| NAS (Zoph & Le, 2017) | 800 | 21-28 |
| Micro NAS (Zoph et al., 2017) | 450 | 3-4 |
| ENAS + general search space | 1 | 0.65 |
| ENAS + search for masks | 1 | 0.48 |
| ENAS + search for skip connections | 1 | 0.52 |
| ENAS + search for cells | 1 | 0.48 |

Table 3: Time and resources needed for different architecture search methods to find the good architectures for CIFAR-10.

## 5 CONCLUSION

Neural Architecture Search (NAS) is an important advance that allows faster architecture design for neural networks. However, the computational expense of NAS prevents it from being widely adopted. In this paper, we presented ENAS, an alternative method to NAS, that requires three orders of magnitude less resources×time. The key insight of our method is to share parameters across child models during architecture search. This insight is implemented by having NAS search for a path within a larger model. We demonstrate empirically that the method works well on both CIFAR-10 and Penn Treebank datasets.

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

# Appendices

## A  DETAILS ON PARAMETER SHARING IN AN RNN CELL

We think of the cell at time step $t$ as a block of $N$ computational nodes, indexed by $\mathbf{h}_1^{(t)}$, $\mathbf{h}_2^{(t)}$, ... $\mathbf{h}_N^{(t)}$. Node $\mathbf{h}_1^{(t)}$ receives two inputs: (1) the RNN signal $\mathbf{x}^{(t)}$ at its current time step and (2) the outputs $\mathbf{h}_D^{(t-1)}$ of the cell from the previous time step. Then, the following computations are performed:

$$\mathbf{c}_1^{(t)} \leftarrow \text{sigmoid}\left(\mathbf{x}^{(t)} \cdot \mathbf{W}^{(\mathbf{x},\mathbf{c})} + \mathbf{h}_N^{(t-1)} \cdot, \mathbf{W}_0^{(\mathbf{c})}\right) \tag{2}$$

$$\mathbf{h}_1^{(t)} \leftarrow \mathbf{c}_1^{(t)} \otimes f_1\left(\mathbf{x}^{(t)} \cdot \mathbf{W}^{(\mathbf{x},\mathbf{h})} + \mathbf{h}_N^{(t-1)} \cdot \mathbf{W}_1^{(\mathbf{h})}\right) + (1 - \mathbf{c}_1^{(t)}) \otimes \mathbf{h}_N^{(t-1)}, \tag{3}$$

where $f_1$ is an activation function that the controller will decide. For $\ell = 2, 3, ..., N$, node $\mathbf{h}_\ell$ receives its input from a layer $j_\ell \in \{\mathbf{h}_1, ..., \mathbf{h}_{\ell-1}\}$, which is specified by the controller, and then performs the following computations:

$$\mathbf{c}_\ell^{(t)} \leftarrow \text{sigmoid}\left(\mathbf{h}_{j_\ell}^{(t)} \cdot \mathbf{W}_{\ell,j_\ell}^{(\mathbf{c})}\right) \tag{4}$$

$$\mathbf{h}_\ell^{(t)} \leftarrow \mathbf{c}_\ell^{(t)} \otimes f_\ell\left(\mathbf{h}_{j_\ell}^{(t)} \cdot \mathbf{W}_{\ell,j_\ell}^{(\mathbf{h})}\right) + (1 - \mathbf{c}_\ell^{(t)}) \otimes \mathbf{h}_{j_\ell}^{(t)}. \tag{5}$$

The shared parameters $\omega$ between different recurrent cells thus consist of all the matrices $\mathbf{W}^{(\mathbf{x},\mathbf{c})}$, $\mathbf{W}^{(\mathbf{x},\mathbf{h})}$, $\mathbf{W}^{(\mathbf{c})}_{\ell,j}$, and $\mathbf{W}^{(\mathbf{h})}_{\ell,j}$. The controller decides the connection $j_\ell$ and the activation function $f_\ell$, for each $\ell \in \{2, 3, ..., N\}$. The layers that are never selected by any subsequent layers are averaged and sent to a softmax head, or to higher recurrent layers. As in the case of convolutional models, to stabilize the training of $\omega$, we add a batch normalization layer after the average of the layers that are not selected.

## B    DETAILS FOR CIFAR-10 SEARCH SPACES

### B.1    DETAILS ON SEARCH SPACE 1: CHANNELS

We use a block size of $S = 32$, resulting in $C/S = 256/32 = 8$ blocks per branch per layer. Each branch configuration has its own embedding and softmax head. To elaborate, this means that a time step in the controller RNN that predicts the configuration for any branch should have a softmax matrix of size $H \times (2^{C/S} - 1)$, where $H = 64$ is the hidden dimension of the RNN, and $2^{C/S} - 1 = 255$ is the number of possible binary masks for that branch. Each branch also has an embedding matrix of size $(2^{C/S} - 1) \times H$, from which the row corresponding to the sampled binary mask is selected and sent to the next time step.

Layers 4 and 8 of our 12-layer network are max pooling layers with a kernel size of $2 \times 2$ and a stride of 2, and reduce each spatial dimension of the layers' outputs by a factor of 2. Within each group of 3 layers where the spatial dimensions of the layers remain constant, we connect each layer to all layers before it (Huang et al., 2016).

### B.2    DETAILS ON SEARCH SPACE 2: CONNECTIONS

We use $3 \times 3$ convolutions with 48 output channels at all layers. The controller RNN for this search space has the same form as the controller RNN that is depicted in Figure 3, with two modifications. First, each block has only 2 time steps (as opposed to 7 in the general search space): (1) the time step that predicts the mask for the convolution of filter size $3 \times 3$, which we force to always activate all channels; and (2) the anchor time step, which we allow to sample multiple previous indices. Second, our controller is thus allowed to form skip connections between arbitrary layers, but forming such connections between layers with different spatial dimensions would result in compilation failures. To circumvent, after each max pooling in the network, we centrally pad the output so that its spatial dimensions remain unchanged.

### B.3    DETAILS ON SEARCH SPACE 3: CONVOLUTION AND REDUCTION CELLS

We perform a $3 \times 3$ convolution on the image to create the outputs for the first convolutional cell. After that, in the first 6 convolution cells, each separable convolution has 32 output channels. Each reduction cell is applied in the same way as the convolutional cell, with the only modification being each operation is applied with a stride of 2. When the cells are found, the final models (both with and without CutOut) are trained for 310 epochs, using the cosine learning schedule, where the reset cycle is originally set to 10 and then doubled after each reset. Additionally, we insert an auxiliary head at the layer immediately before the second application of the reduction cell (Szegedy et al., 2016).

