# OpenReview forum: "Faster Discovery of Neural Architectures by Searching for Paths in a Large Model"
_ICLR.cc/2018/Conference — Invite to Workshop Track_

### Official Review · AnonReviewer1 · 2017-11-26
**an incremental work, but the results are not bad**

**Rating:** 6
**Confidence:** 2

**Review:**

In the paper titled "Faster Discovery of Neural Architectures by Searching for Paths in a Large Model", the authors proposed an efficient algorithm which can be used for efficient (less resources and time) and faster architecture design for neural networks. The motivation of the new algorithm is by sharing parameters across child models in the searching of archtecture. The new algorithm is empirically evaluated on two datasets (CIFAR-10 and Penn Treeback) --- the new algorithm is 10 times faster and requires only 1/100 resources, and the performance gets worse only slightly.

Overall, the paper is well-written. Although the methodology within the paper appears to be incremental over previous NAS method, the efficiency got improved quite significantly.

---

> ### Author Response · Authors · 2017-12-28
> **Rebuttal for AnonReviewer1**
>
> We thank the reviewer for their review. To address reviewers’ concerns, we have completely rewritten the paper to make it easier to follow and we have also included new experimental results (all reported in our revision). We sincerely hope that the reviewer will update positively on our revised paper. We believe that ENAS delivers extremely non-trivial contributions to architecture search approaches.
>
> First, while the idea of ENAS is simple - sharing parameters between architectures so that they don’t need to be trained again - we believe that this idea is far from incremental. A lot of details are needed to share the parameters appropriately, e.g. the design choice of representing all child models in a large graph (see Sections 3 and 4 in our revision for more details). ENAS leads to two orders of magnitude reduction of computing resource and an order of magnitude reduction of time consumed, all without sacrificing much performance.
>
> Second, in our revision, we showed that ENAS actually achieves a better perplexity than NAS on Penn Treebank. We designed a different search space, in which ENAS found a recurrent cell that achieves 57.8 perplexity on Penn Treebank (compared to NAS’s 62.4 perplexity), and establishes the state-of-the-art performance on automatic model design on Penn Treebank. Furthermore, this result is also achieved without extensive hyper-parameters tuning (Melis et al., 2017). Details are in Section 4.1 of our revision.
>
> Finally, we find that among other submissions to ICLR 2018, some papers address the similar problem with ENAS: the computational expense of automatic model designing approaches. For example SMASH [1] is a method that automatically designs networks for image classification tasks. As of now, despite the fact that SMASH performs worse than ENAS and was only applied to images and not texts, their paper has a much better score than ours (SMASH has an averaged score of 6 while our paper got an averaged score of 5). We sincerely hope that you will give ENAS another consideration.
>
> [1] Paper 1: SMASH: One-Shot Model Architecture Search through HyperNetworks (https://openreview.net/forum?id=rydeCEhs-)

---

### Official Review · AnonReviewer3 · 2017-11-26
**Nice idea, but the paper suffers from unclear presentation and the empirical results are not convincing enough.**

**Rating:** 5
**Confidence:** 3

**Review:**

Summary:
The paper presents a method for learning certain aspects of a neural network architecture, specifically the number of output maps in certain connections and the existence of skip connections. The method is relatively efficient, since it searches in a space of similar architectures, and uses weights sharing between the tested models to avoid optimization of each model from scratch. Results are presented for image classification on Cifar 10 and for language modeling.

Page 3: “for each channel, we only predict C/S binary masks”   -> this seems to be a mistake. Probably “for each operation, we only predict C/S binary masks” is the right wording
Page 4: Stabilizing Stochastic Skip Connections: it seems that the suggested configuration does not enable an identity path, which was found very beneficial in (He. et al., 2016). Identity path does not exist since layers are concatenated and go through 1*1 conv, which does not enable plain identity unless learned by the 1*1 conv.
Page 5:
-	The last paragraph in section 4.2 is not clear to me. What does a compilation failure mean in this context and why does it occur? And: if each layer is connected to all its previous layers by skip connections, what remains to be learned w.r.t the model structure? Isn’t the pattern of skip connection the thing we would like to learn?
-	Some  details of the policy LSTM network are also not clear to me:
o	How is the integer mask (output of the B channel steps) encoded? Using 1-hot encoding over 2^{C/S} output neurons? Or maybe C/S output neurons, used for sampling the C/S bits of the mask? this should be reported in some detail.
o	How is the mask converted to an input embedding for the next step? Is it by linear multiplication with a matrix? Something more complicated? And are there different matrices used/trained for each mask embedding (one for 1*1 conv, one for 3*3 conv, etc..)?
o	What is the motivation for using equation 5 for the sampling of skip connection flags? What is the motivation for averaging the winning anchors as the average embedding for the next stage (to let it ‘know’ who is connected to the previous?). Is anchor j also added to the average?
o	How is equation 5 normalized? That is: the probability is stated to be proportional to an exponent of an inner product, but it is not clear what the constant is and how sampling is done.

Page 6:
-	  Section 4.4: what is the fixed policy used for generating models in the stage of training the shared W parameters? (this is answered at page 7
Experiments:
-	The accuracy figures obtained are impressive, but I’m not convinced the ENAS learning is the important ingredient in obtaining them (rather than a very good baseline)
-	Specifically, in the Cifar -10 example it does not seem that the networks chooses the number of maps in a way which is diverse or different from layer to layer. Therefore we do not have any evidence that the LSTM controller has learnt any interesting rule regarding block type, or relation between block type width and layer index. All we see is that the model does not chose too many maps, thus avoid significant overfit. The relevant baseline here is a model with 64 or 96 maps on each block, each layer.Such a model is likely to do as well as the ENAS model, and can be obtained easily with slight parameter tuning of a single parameter.
-	 Similarly, I’m not convinced the  skip connection pattern found for Cifar-10 is superior to standard denseNet or Resnet pattern. The found configuration was not compared to these baselines. So again we do not have evidence showing the merit of keeping and tuning many parameters with the RINFORCE
-	The experiments with Penn Treebank are described with too less details: for example, what exactly is the task considered (in terms on input-output mapping), what is the dataset size, etc..
-	Also, for the Penn treebank experiments no baseline is given, so it is not possible to understand if the structure learning here is beneficial. Comparison of the results to an architecture with all skip connections, and with a single skip connection per layer is required to estimate if useful structure is being learnt.

Overall:
-	Pro: the method gives high accuracy results
-	Cons:
o	It is not clear if the ENAS search is responsible to the results, or just the strong baseline. The advantage of ENAS over plain hyper parameter choosing was not sufficiently established.
o	The controller was not presented in a clear enough manner. Many of its details stay obscure.
o	The method does not seem to be general. It seems to be limited to choosing a specific set of parameters in very specific scenario (scenario which enable parameter sharing between model. The conditions for this to happen seems to be rather strict, and where not elaborated).

After revision:
The controller is now better presented.
However, the main points were not changed:
   - ENAS seems to be limited to a specific architecture and search space, in which probably the search is already exhausted. For example for the image processing network, it is determining the number of skip connections and structure of a single layer as a combination of several function types. We already know the answers to these search problems (denser skip connection pattern works better, more functions types  in a layer in parallel do better, the number of maps should be adjusted to the complexity and data size to avoid overfit). ENAS does not reveal a new surprising architectures, and it seems that instead of searching in the large space it suggests, one can just tune a 1-2 parameters  (for the image network, it is the number of maps in a layer).
  - Results comparing ENAS results to the simple baseline of just tuning 1-2 hyper parameters were not shown. I hence believe the strong empirical results of ENAS are a property of the search space (the architecture used) and not of the search algorithm.

---

> ### Public Comment · (anonymous) · 2017-12-05
> **Comparison with baseline model**
>
> Doesn't the result of Sanity Check with Ablation Study section imply that REINFORCE successfully learned a competitive architecture from all the possible ones? I think the resulting performance being not much better than that of the baseline is because of the chosen search space. If they adapted the search space of "Learning Transferable ..." by Zoph et. al., they would be able to achieve a comparable performance given they used PPO instead, since they achieved the performance comparable to that of NAS by Zoph & Le. I think it's bit too harsh to give 4 for a paper that reduced the computation cost of NAS to 1/100. SMASH achieved much higher score from the reviewers, but they achieved a similar performance. It relies on parameter sharing assumption as well, and they demonstrated that the assumption is valid in their case and therefore reasonable to assume for similar cases. Since the author of ENAS cited SMASH paper, I don't think they have to mention the assumption. You claim that many of the details of the controller are obscure, but we, a third party, didn't experience much difficulty in implementing this algorithm for CNN part after asking a few questions below. So, I'd argue that just a few are obscure, which happens among successful papers as well.

---

> > ### Author Response · Authors · 2017-12-28
> > **Thank you for the nice words!**
> >
> > We thank the Anonymous poster of this comment.

---

> ### Author Response · Authors · 2017-12-28
> **Rebuttal for AnonReviewer3**
>
> We thank the reviewer for the comments. We were very dismayed by the low score. Subsequently, we have completely rewritten the paper to make it easier to follow .We have also included new experimental results to address the reviewer’s concerns. All the results are reported in our revision. We hope that our revisions of the paper can clear the reviewer’s reservation about ENAS’s ability.
>
> In the following, we try to address the reviewer’s concerns.
>
> First, we have completely rewritten the paper for clarity. We focused on delivering the high level ideas, and we moved a lot of implementation details into our appendix. To address the reviewer’s particular comment about the lack of details on the controller, we refer the reviewer to Section 3 in our revision.
>
> As evidenced by the anonymous comment above, our previous presentation is sufficient for independent researchers to to implement our method. Therefore, we believe that our revision, which aims to improve the original presentation, does not obscure ENAS’s details. After the reviewing cycle, we will also publish our code, which we hope will clear any ambiguity about ENAS.
>
> Second, in our revision, we have included new experimental results to address the reviewer’s concerns that ENAS is not general, and whether ENAS is responsible for the good results. We summarize them below:
>
> 1) ENAS is indeed responsible for the results. This information was in our original submission. In our revision, it is highlighted in the paragraph “Sanity Check and Ablation Study” at the beginning Section 4.3. In particular, a model randomly sampled from our search space does not perform as well as a model sampled by the ENAS controller. Also, we if train ENAS without training its controller, performance is much worse. Both observations, as presented in the paper, indicate the importance of ENAS.
>
> To further address the reviewer’s concerns, we have conducted more controlled experiments. Following are their results:
>
> 1a) 64 or 96 maps on CIFAR-10 models. Sure, a model with randomly chosen 64 or 96 maps on each block, each layer may perform similarly to ENAS. However, in this search space, the controller can take up to 256 maps. Without ENAS, a random model designer would select roughly 128 maps at each block, each layer. If you haven’t seen ENAS’s decisions to pick 64 or 96 maps, would you think of such a baseline? We do agree that a slight tune of hyper-parameters may also lead to this model. However, in other search spaces (e.g. see Section 4.2 in our revision), where one needs to figure out the skip connections, the tuning of hyper-parameters won’t be as “slight”.
>
> 1b) The pattern of skip connections found by ENAS is indeed better than the DenseNet pattern and the ResNet pattern. In our settings, the DenseNet pattern (connecting every pair of layers) achieves 5.23% test error, and the ResNet pattern (connecting each layer to the next) achieves 6.01% test error. We also note that the DenseNet and the ResNet patterns in our settings are not the DenseNet and ResNet in their original papers. The reason for the differences lies in the design choice of our search spaces: we make skip connections go through a conv1x1 instead of concatenation as in the original DenseNet (Huang et al., 2016), or identity and addition as in the original ResNet (He et al., 2015). Such design choice may be sub-optimal, and we will try the identity skip connections in our next revision. However, our controlled experiment does show that in our search space, the skip connections that ENAS finds do achieve non-trivial improvements compared to standard patterns.
>
> 2) ENAS is a general method. To see this, note that one way to do programming is to search for a path in a bigger program, where all operations are available at every step. In ENAS, the computations in a neural architecture can be viewed as a program, which is represented as directed acyclic graph (DAG) (see Section 1 of our revision). To apply ENAS to any task, e.g. designing a convolutional network, or designing a recurrent cell, one only needs to specify the DAG’s components (examples are now in Section 3 of our revision).
>
> 3) We further elaborate point 2) above by applying ENAS to different search spaces. First, we use ENAS to search for both skip connections and layer operations (convolutions with different filter sizes, or average pooling, or max pooling). It turns out that in this search space, ENAS could discover a model with CIFAR-10 test error of 4.23%. This resulting model is comparable to the model found in the restricted search space over convolutions and pooling masks. Therefore, ENAS works in this search space.
>
> [to be continued]

---

> > ### Author Response · Authors · 2017-12-28
> > **Rebuttal for AnonReviewer3 (continued 1)**
> >
> > 4) More details on Penn Treebank experiments have been added. We designed a different search space for recurrent cells, in which ENAS finds a novel recurrent cell that achieves 57.8 test perplexity on Penn Treebank. We have reported this new result in our revision (Sections 4.1 and 5.1). Let’s call this the ENASCell.
> >
> > ENASCell is very novel compared to the recurrent highway network (see Figure 4 in our revision). While the search space for ENASCell still uses highway connections, the ENAS controller has discovered several novelties:
> > - the use of the ReLU activation, unlike in recurrent highway network where only the tanh activation is used
> > - the pattern of connections within the ENASCell
> > We have also mentioned in the revision that ENASCell is, in a sense, a local optimum. If we slightly vary its components, its performance drops. This means that ENASCell is not trivial to find, affirming the role of ENAS.
> >
> > Additionally, to our knowledge, ENASCell’s perplexity of 57.8 is the state-of-the-art among automatic model design approaches on Penn Treebank. It outperforms NAS (62.4 perplexity), which uses two orders of magnitude more computations and way more time. ENASCell, with almost no hyper-parameter tuning, also outperforms LSTM with extensive hyper-parameters tuning (59.5 perplexity) (Melis et al., 2017).
> >
> > Based on these results, we believe that it is clear that: 1) ENAS a crucial component to our design of novel architectures that achieves good performances and 2) ENAS is a general method: whenever a search space is specified, ENAS is applicable. ENAS’s performance indeed depends on the search space, but this is also the case with other NAS methods.
> >
> > Finally, we can compare ENAS to other ICLR 2018 submissions that address the computational expense of automatic model designing approaches. SMASH [1] is a method that automatically designs networks for image classification tasks. As of now, despite the fact that SMASH performs worse than ENAS and was only applied to images and not texts, their paper has an average score of 6 while our paper received an averaged score of 5.
> >
> > We believe that ENAS delivers significant contributions to automatic model designing, and that ENAS has compelling advantages compared to hyperparameter tuning, especially in its ability to achieve good performances with a low usage of computing resource and time. We sincerely hope that you will give ENAS a reconsideration.
> >
> > [1] Paper 1: SMASH: One-Shot Model Architecture Search through HyperNetworks (https://openreview.net/forum?id=rydeCEhs-)
> > ----------------------------------------

---

> > > ### Author Response · Authors · 2017-12-28
> > > **Rebuttal for AnonReviewer3 (continued 2, end)**
> > >
> > > While we hope that our revision will improve your overall understanding of the paper, we will answer your specific questions below:
> > >
> > > Page 3: “for each channel, we only predict C/S binary masks”   -> this seems to be a mistake. Probably “for each operation, we only predict C/S binary masks” is the right wording.
> > > => Thank you. This is indeed the incorrect wording.  We have presented this search space differently in the revision.
> > >
> > > Page 4: Stabilizing Stochastic Skip Connections: it seems that the suggested configuration does not enable an identity path, which was found very beneficial in (He. et al., 2016). Identity path does not exist since layers are concatenated and go through 1*1 conv, which does not enable plain identity unless learned by the 1*1 conv.
> > > => Indeed, in all ENAS search spaces presented in the paper, there is no identity path. When we designed ENAS, we thought that since ENAS allows each layer to be sent stochastically to any layer above, each layer should at least go through a different transformation in its skip connections. However, thanks to your comment, we believe that this requirement can be relaxed. We will experiment with identity skip connections in the final revision of the paper.
> > >
> > > Page 5:
> > > -	The last paragraph in section 4.2 is not clear to me. What does a compilation failure mean in this context and why does it occur? And: if each layer is connected to all its previous layers by skip connections, what remains to be learned w.r.t the model structure? Isn’t the pattern of skip connection the thing we would like to learn?
> > > => We present this part differently in the revision. Please refer to the paragraph “Search Spaces” in Section 4.2 of the revision.
> > >
> > > -	Some  details of the policy LSTM network are also not clear to me:
> > > o	How is the integer mask (output of the B channel steps) encoded? Using 1-hot encoding over 2^{C/S} output neurons? Or maybe C/S output neurons, used for sampling the C/S bits of the mask? this should be reported in some detail.
> > > => Each mask is an integer between 1 and 2^(C/S) - 1, and is encoded using one-hot encoding.
> > > o	How is the mask converted to an input embedding for the next step? Is it by linear multiplication with a matrix? Something more complicated? And are there different matrices used/trained for each mask embedding (one for 1*1 conv, one for 3*3 conv, etc..)?
> > > => Each mask has its own embedding. If there are B x (2^(C/S) - 1) possible masks (one for each operation), then there will be B x (2^(C/S) - 1) embedding vectors.
> > > o	What is the motivation for using equation 5 for the sampling of skip connection flags? What is the motivation for averaging the winning anchors as the average embedding for the next stage (to let it ‘know’ who is connected to the previous?). Is anchor j also added to the average?
> > > => The idea of using attention weights to sample skip connections is inspired from Zoph and Le (2017). In their paper, Zoph and Le sampled each connection using a Bernoulli. In ENAS, we sample multiple connections using a multinomial. We average the winning anchors to tell the controller LSTM which previous layers have been sampled. Anchor j is not added to the average.
> > > o	How is equation 5 normalized? That is: the probability is stated to be proportional to an exponent of an inner product, but it is not clear what the constant is and how sampling is done.
> > > => The probabilities are normalized by the sum of exp(.) for all previous steps.

---

> ### Author Response · Authors · 2018-01-03
> **Let's discuss your remaining concerns!**
>
> We thank the reviewer for reading our rebuttals, recognizing that the controller has been better presented, and updating the score. However, we are still unsatisfied with the current evaluation of our work, especially based on the remaining concerns that the reviewer has raised.  We have supporting evidence to completely address the concerns raised by the reviewer.
>
> *** Note that the paper has been updated with a new / much cleaner search space and better results on the PTB dataset. See Section 3.1 and 4.1. We achieved the perplexity of 57.8, much better than NAS’s previous result of 62.4, and the recurrent cell that ENAS found cannot be obtained with hyper-parameters tuning. We suspect that the reviewer did not see these updates. We use these results in the comments below to address some concerns by the reviewer. We encourage the reviewer to take a look at these sections before reading the rebuttal below.
>
> 1. The reviewer comments “ENAS seems to be limited to a specific architecture and search space, in which probably the search is already exhausted.”
>
> We argue that ENAS and NAS are fundamentally equivalent. Searching a path within a model and searching different operation per step are the same.
>
> We disagree with this statement “the search is already exhausted”. As presented in Section 3.1, the size of the search space for recurrent cells is about 8.03 × 10^15. ENAS has only seen 735,000 architectures, which is very far away from exhausting the search space (1 in 10^10). We are likewise only able to search for 759500 architectures of our convolutional search space, where the search space size is about 1.6 × 10^29.
>
> 2. The reviewer comments “For example for the image processing network, it is determining the number of skip connections and structure of a single layer as a combination of several function types.”
>
> This observation is also not true. ENAS does not only find “the number of skip connections” but also finds what are those skip connections. It is very clear from Figure 5-Right in our revision. There are 26 skip connections. If one is told that a network with 12 layers needs 26 skip connections, there are still (12C2)C26 = 1.65 × 10^18 possible choices.
>
> 3. The reviewer comments “We already know the answers to these search problems (denser skip connection pattern works better,”
>
> We disagree with this intuition and we have the evidence to support our disagreement. Denser skip connection patterns require more parameters (for conv 1x1), which may lead to overfitting. Indeed, we did a controlled experiment, and the DenseNet pattern (connecting every pair of layers) achieves 5.23% test error, which is worse than the pattern of skip connections found by ENAS. (Note: we have mentioned this in our first round of rebuttal comments)
>
> 4. The reviewer comments “ENAS does not reveal a new surprising architectures, and it seems that instead of searching in the large space it suggests, one can just tune a 1-2 parameters (for the image network, it is the number of maps in a layer). ... Results comparing ENAS results to the simple baseline of just tuning 1-2 hyper parameters were not shown.”
>
> The reviewer is concerned that ENAS didn’t find any surprising architectures. We disagree and argue that the recurrent cell is surprising and cannot easily designed manually  (Figure 4 in our revision). It is subjective but one can also argue that the architectures on CIFAR-10 aren’t obvious.
>
> In terms of hyperparameter tuning vs architecture search, we have the following pieces of evidence:
>
> First, we tuned the number of maps at each layer of the DenseNet pattern (64, 128, and 256). The best test error we could get was 4.07%. Second, we randomly sampled a pattern of skip connections, and then tuned the number of maps at each layer (we tried 48, 64, 128, 256, and 512). The lowest test error we could get by doing so is 5.11%. Both of these test errors are worse than the 3.87% obtained by ENAS. We’re happy to add these results to the paper.
>
> Thirdly, Melis et al (2017) [1] has performed extensive tuning of an LSTM network. Zoph and Le (2017) also reported to have done a grid search over hyper-parameters. Both of them used intensive computing resources to tune way more than 1-2 hyper-parameters, and yet neither achieved a performance as good as our ENAS recurrent cell (58.9 and 62.4 perplexity, compared to 57.8 by ENAS), and we haven’t tuned any hyper-parameters! It is thus clear that hyper-parameters tuning will not lead to comparable performance with ENAS, at least not without a good model.
>
> [to be continued]

---

> > ### Author Response · Authors · 2018-01-03
> > **Let's discuss your remaining concerns! (continued and end)**
> >
> > 5. The reviewer comments “I hence believe the strong empirical results of ENAS are a property of the search space (the architecture used) and not of the search algorithm.”
> >
> > We agree with this statement “the strong empirical results of ENAS are a property of the search space”. However, the performance of *any* NAS algorithm depends on the search space. Please see the improvements in results from the original NAS paper [2] to the latest NAS paper [3] (state-of-art in CIFAR10, ImageNet), where the change is mainly in the search space. To quote their abstract in [3] “Our key contribution is the design of a new search space which enables transferability.”
> >
> > We disagree with your comment that the strong results of NAS is not due to the search algorithm. Here’s some supporting evidence:
> >
> > - Section 4.1, just above Table 1. The recurrent cell that ENAS finds does not have identity or sigmoid activations, while they are available in the search space. ENAS learns to ignore them. Furthermore, random perturbations in the ENAS recurrent cell worsen the result.
> > - Section 4.3, just above Table 3. Without properly training the search algorithm of ENAS, one cannot find a good network architecture. In fact, an independent researcher has commented below that “the result of Sanity Check with Ablation Study section imply that REINFORCE successfully learned a competitive architecture from all the possible ones”.
> >
> > [1] Gabor Melis, Chris Dyer, and Phil Blunsom. On the state of the art of evaluation in neural language models. Arxiv, 1707.05589, 2017.
> > [2] Barret Zoph, Quoc V. Le. Neural Architecture Search with Reinforcement Learning. ICLR, 2017.
> > [3] Barret Zoph, Vijay Vasudevan, Jonathon Shlens, Quoc V. Le. Learning Transferable Architectures for Scalable Image Recognition. arXiv, 1707.07012, 2017.

---

> > ### Comment · AnonReviewer3 · 2018-01-03
> > **Short reply to 'remaining concerns discussion'**
> >
> > I  am here briefly referring to the author's points:
> >
> > 1) The claim is not that the search space is small and we have visited every point of it. Instead it is that it is not very 'interesting' in the sense that most of the architectures in it are similar to each other and the accuracy in this subspace is actually determined by a small number of hyper parameters (number of maps, density of skip connections). The search is hence in an over parametrized space which is just seemingly large
> > 2) This is the same point as 1). It is clear to me that ENAS looks for the pattern of skip connection and not only their number. However, I'm not convinced that the pattern is very important. and it seems that the number and maybe 1 additional parameter (whether connection should be focused on lower or upper layers) are important.
> > 3,4) the results of standard dense net with varying number of maps per layer are very important, as they provide a real relevant comparison baseline for ENAS. However, 1) these results were not, and are still not presented in the paper, and 2) looking at then, I believe they support my intuition rather than disprove it: it seems that most of the accuracy gained by ENAS (4.07  vs. 3.87) can be gained from searching in a 1-D search space: Dense net with varying number of maps per layer.
> > Regarding the LSTM accuracy obtained: I'm less familiar with SOA in text modeling cells, so it is hard for me to judge the cell found by ENAS and its ingenuity.
> >
> > 5) Indeed the success of a NAS algorithm depends on the quality of the search space. ENAS is able to obtain good results because its search space includes state of the art architectures. However, as stated above, as far as I can see, the search space formulated is heavily over parametrized: it tries to tune many parameters with huge number of possibilities, where a much smaller space of 2-3 dimensions to tune would have obtained almost the same results. Results checking this view (for example, the fact that tuning a single parameter (the number of maps in a dense net) gives almost all the accuracy obtained by ENAS) are not clearly presented in the paper. As far as I can see, ENAS formulates a too-complicated search space and does not seem to benefit from it a lot.

---

> > > ### Author Response · Authors · 2018-01-04
> > > **Further discussions: general search space, and significance of results**
> > >
> > > The reviewer has many concerns, but we believe that the reviewer is not impressed by the search space, and the fact that the search space is not interesting. We have a result in the paper where we ran ENAS with a general search space, and search for both skip connection patterns and operations at each layer (Section 4.2, last paragraph of page 7). This search space is as general as the search space in the original NAS paper [1], and is 16M larger than the constrained search spaces. ENAS found a model that achieves 4.23% test error. This result is on par with one of the best human-designed architectures in 2016: WideResNet (4.17% test error).
> > >
> > > Even within the restricted search space over patterns skip connections, the patterns in the subspace only **look** similar. They have a wide range of accuracy: a randomly chosen pattern of skip connections has test error 5.11% (our previous comment); densenet pattern has test error 4.07%; the best pattern that ENAS finds has test error 3.87%. The relative improvement compared to the random baseline is (ENAS - densenet) / (ENAS - random) = 0.16, which is statistically significant.
> > >
> > > We understand that the reviewer may not understand the significance of the new recurrent cells and 57.8 perplexity. Here we want to explain its significance: Recurrent Highway Networks [2] was accepted at ICML 2017 by making an improvement of 3 perplexity on a strong baseline, going from 68.5 to 65.4. Their paper is cited 72 times within a year. Here, we are making a similar improvement of 4.6 perplexity on a much stronger baseline, going from 62.4 to 57.8, and setting the state-of-the-art among automatic model design methods.
> > >
> > > [1] Barret Zoph, Quoc V. Le. Neural Architecture Search with Reinforcement Learning. In ICLR, 2017.
> > > [2] Julian Georg Zilly, Rupesh Kumar Srivastava, Jan Koutnik, and Jurgen Schmidhuber. Recurrent highway networks. In ICML, 2017.

---

### Official Review · AnonReviewer2 · 2017-11-27
**Improving Neural Architecture Search by Parameter Sharing**

**Rating:** 5
**Confidence:** 2

**Review:**

In this paper, the authors look to improve Neural Architecture Search (NAS), which has been successfully applied to discovering successful neural network architectures, albeit requiring many computational resources. The authors propose a new approach they call Efficient Neural Architecture Search (ENAS), whose key insight is parameter sharing. In NAS, the practitioners have to retrain for every new architecture in the search process, but in ENAS this problem is avoided by sharing parameters and using discrete masks. In both approaches, reinforcement learning is used to  learn a policy that maximizes the expected reward of some validation set metric. Since we can encode a neural network as a sequence, the policy can be parameterized as an RNN where every step of the sequence corresponds to an architectural choice. In their experiments, ENAS achieves test set metrics that are almost as good as NAS, yet require significantly less computational resources and time.

The authors present two ENAS models: one for CNNs, and another for RNNs. Initially it seems like the controller can choose any of B operations in a fixed number of layers along with choosing to turn on or off ay pair of skip connections. However, in practice we see that the search space for modeling both skip connections and choosing convolutional sizes is too large, so the authors use only one restriction to reduce the size of the state space. This is a limitation, as the model space is not as flexible as one would desire in a discovery task. Moreover, their best results (and those they choose to report in the abstract) are due to fixing 4 parallel branches at every layer combined with a 1 x 1 convolution, and using ENAS to learn the skip connections. Thus, they are essentially learning the skip connections while using a human-selected model.

ENAS for RNNs is similar: while NAS searches for a new architecture, the authors use a recurrent highway network for each cell and use ENAS to find the skip connections. Thus, it seems like the term Efficient Neural Architecture Search promises too much since in both tasks they are essentially only using the controller to find skip connections. Although finding an appropriate architecture for skip connections is an important task, finding an efficient method to structure RNN cells seems like a significantly more important goal.

Overall, the paper is well-written, and it brings up an important idea: that parameter sharing is important for discovery tasks so we can avoid re-training for every new architecture in the search process. Moreover, using binary masks to control network path (essentially corresponding to training different models) is a neat idea. It is also impressive how much faster their model performs on tasks without sacrificing much performance. The main limitation is that the best architectures as currently described are less about discovery and more about human input -- finding a more efficient search path would be an important next step.

---

> ### Author Response · Authors · 2017-12-28
> **Rebuttal for AnonReviewer2**
>
> We thank the reviewer for the comments. We were very dismayed by the low score. Subsequently, we have completely rewritten the paper to make it easier to follow. We have also included new experimental results to address the reviewer’s concerns. All the results are reported in our revision and are summarized below. We hope that our revisions of the paper can clear the reviewer’s reservation about ENAS’s ability.
>
> Summary of New Results:
>
> 1) The reviewer is concerned that ENAS can only search small search spaces. This is not the case. We used ENAS to search for both skip connections and layer operations (convolutions with different filter sizes, or average pooling, or max pooling). It turns out that in this large search space, ENAS could also discover a model with CIFAR-10 test error of 4.23%. This resulting model is comparable to the model found in the restricted search space over convolutions and pooling masks. Therefore, search space size is not a limitation of ENAS.
>
> 2) The reviewer is concerned that the best ENAS model is “less about discovery and more about human input.” We show that ENAS can do well with less human inputs. In particular, we took the pattern of skip connections discovered by ENAS (Figure 5-Right in our revision), and simply increased the number of output channels at each layer from 256 to 512. The resulting model achieves 3.87% test error on CIFAR-10. In the original paper, the best ENAS result on CIFAR-10 is 3.86% test error, achieved by using multiple branches at each layer. Therefore, we showed that the model found by ENAS, with minimal human inputs, can achieve a similar performance to models that are designed with more human inputs.
>
> We note that ENAS’s principle, i.e. searching for a path in a big model is general. Under this principle, we can do whatever other NAS approaches can do. We also note that the human input of increasing the number of output channels was also performed by the original NAS paper (Zoph and Le, 2017).
>
> 3) We designed a different search space for recurrent cells. In this search space, ENAS finds a novel recurrent cell that achieves 57.8 test perplexity on Penn Treebank. We have reported this new result in our revision (Sections 4.1 and 5.1). Let’s call this the ENASCell. ENASCell is very novel compared to recurrent highway network (see Figure 4 in our revision). While the search space for ENASCell still benefits from highway connections, the ENAS controller has discovered several novelties:
> - the use of the ReLU activation, unlike in recurrent highway network where only the tanh activation is used
> - the pattern of connections within the ENASCell
> To our knowledge, ENASCell’s perplexity of 57.8 is the state-of-the-art among automatic model design approaches on Penn Treebank. ENASCell outperforms NAS (62.4 perplexity), which uses two orders of magnitude more computations and one order of magnitude more time. ENASCell, with almost no hyper-parameters tuning, also outperformed LSTM with extensive hyper-parameters tuning (59.5 perplexity) (Melis et al., 2017).
>
> We now compare ENAS to other ICLR 2018 submissions which address similar problems, i.e. computationally inexpensive approaches for automatic model designing. In particular, Paper 1 presents SMASH, a method that automatically designs networks for image classification tasks, and Paper 323 presents a method that automatically designs recurrent cells. As of now, despite the fact that their methods perform worse than ENAS, both papers receive at least one 7 in their reviews.
>
> We believe that ENAS’s contributions are significant, both in the novelty of its idea and in the significance of its results. We sincerely hope that you will give ENAS another consideration.
>
>
> [1] Paper 1: SMASH: One-Shot Model Architecture Search through HyperNetworks (https://openreview.net/forum?id=rydeCEhs-)
> [2] Paper 323: A Flexible Approach to Automated RNN Architecture Generation (https://openreview.net/forum?id=SkOb1Fl0Z)

---

### Public Comment · (anonymous) · 2017-11-07
**Is M=1 OK for training policy, too?**

1) Was there any empirical result that made you choose REINFORCE rule instead of using PPO loss function (as in "Learning Transferable Architectures for Scalable Image Recognition") for updating theta, which was stated to work better in Zoph et. al. 2017?

2) In the section "training shared parameter omega," it says M=1 is sufficient. Does this apply to "Training the Policy π" as well?

---

> ### Author Response · Authors · 2017-11-08
> **M=1 doesn't work for training policy**
>
> 1. We went with REINFORCE for the ease of implementation. We have not tried other methods, such as PPO, TRPO, etc. but we will look into this soon.
>
> 2. We tried, and we needed at least M=10 to training the policy π. We suspect this is because the gradient estimated with REINFORCE has a high variance. M=1 works for training omega because every update for omega has its gradient computed on a batch of training example, leading to a smaller variance.

---

### Public Comment · (anonymous) · 2017-11-21
**What is the input to each timestep of the controller RNN?**

In Figure 2 of ENAS, what exactly is the input (besides the RNN hidden state) to each timestep of the RNN?

Does the mask index a row of a different input embedding matrix (1 of 256 rows in 1 of 7 possible 256x64 matrices) (256 is from 2^(C/S)-1) (7 is from the 7 possible layer components) depending on whether the mask from previous timestep output was used to mask a 1x1_conv, 3x3_conv, 5x5_conv, 7x7_conv, max_p, avg_p, or skip_anchor? <--Is this row embedding method the only/correct input to the RNN or is it actually something else?

---

> ### Author Response · Authors · 2017-11-21
> **Sharing mask embeddings between operations**
>
> Thanks for the question.
>
> The input to each time step of the controller RNN at time step t is the embedding of the decision sampled from time step t-1. As there are 2^(C/S)-1 possible decisions at each time step, the embedding matrix has 2^(C/S)-1 rows, which are shared among all 6 operations (1x1_conv, 3x3_conv, 5x5_conv, 7x7_conv, max_p, and avg_p). So to answer your second question, the mask indices from the same embedding matrix are shared among these 6 operations.
>
> Meanwhile, the skip_anchor is treated differently. The input it provides to the next step (which decides the mask for a 1x1_conv) is the mean of the previous anchor steps that get sampled. We describe this in the paragraph right below Equation (5) in the paper.

---

### Public Comment · (anonymous) · 2017-12-01
**Did you try updating omega and theta alternately for each iteration rather than for each epoch?**

After an epoch of omega updates, theta updates were performed. Did you alternate more frequently to see what happens?

What if you replace the reward with loss function or some combination of loss function and classification accuracy?

If you apply ENAS longer, say for 3 days, will the resulting performance be better? Did you find the performance to converge around the 300 epochs?

After the 300 epochs, you sampled many architectures and determined the one with the highest reward to be one trained from scratch. How many architecture samples did you find to be enough (for convergence)?

---

> ### Author Response · Authors · 2017-12-28
> **We didn't try try updating omega and theta alternately for each iteration rather than for each epoch**
>
> | Did you alternate more frequently to see what happens?
>
> We haven't try alternating more often. We suspect that this will make little change to CIFAR-10, but will perhaps change the results for PTB. This is because in PTB, one carries over the last RNN stage.
>
> | What if you replace the reward with loss function or some combination of loss function and classification accuracy?
>
> We haven't tried this. However, there is a reason that we chose the reward as in the paper. If you look at the appendix in our original submission, you'd see that those reward functions are chosen to in a similar way to Expectation-Maximization updates: they lead to a surrogate to our objective.
>
> | If you apply ENAS longer, say for 3 days, will the resulting performance be better? Did you find the performance to converge around the 300 epochs?
>
> We suspect there will be no longer improvement, as we observed convergence in the controller's samples small entropy. However, further improvements are possible, if one comes up with an alternative search space. This is the case for the new Penn Treebank results in our revision (57.8 perplexity).
>
> | After the 300 epochs, you sampled many architectures and determined the one with the highest reward to be one trained from scratch. How many architecture samples did you find to be enough (for convergence)?
>
> In our 300 epochs, about 300 * 2000 = 600,000 architectures were sampled. We didn't keep track of whether these architectures are different, and we suspect that towards the end of the training process, the sampled architectures are very similar to each other.

---

### Author Response · Authors · 2017-12-28
**Revision and General rebuttal**

We thank the reviewers for their comments.

A general concern shared among the reviewers is that the paper is not clear. To address this concern, we have completely rewritten the paper. Please see our most recent revision for the changes. We did so with two main goals:

1. Make the presentation of ENAS easier to follow. We tried to present examples of how ENAS and its controller work (Section 3 in our revision), and moved the detailed implementations to our Experiments and Appendix. We will eventually release our implementation of ENAS to clear all ambiguities for readers who wish to replicate our method.

2. Present new experimental results. We conducted experiments on other search spaces, as suggested by the comments of AnonReviewer2 and AnonReviewer3. With these experiments, we hope to show that ENAS is a general method, and that ENAS can discover good models with minimal human inputs (Section 4 in our revision). For details, please see our revision as well as the comments delivered to each reviewer.

We were dismayed by the low scores that we received. Not only does ENAS speed up NAS by an order of magnitude, but in our revision, ENAS also achieves the state-of-the-art performance on Penn Treebank among automatic model design approaches (57.8 perplexity, without extensive hyper-parameters tuning). Other ICLR 2018 submissions, e.g., Paper 1 and Paper 323, also try to address the expensive use of time and computing resource by NAS approaches. They have worse empirical results than ENAS, and yet receive better scores. We thus hope that the reviewers will reconsider their judgement of our work.

References:
[1] Paper 1: SMASH: One-Shot Model Architecture Search through HyperNetworks (https://openreview.net/forum?id=rydeCEhs-)
[2] Paper 323: A Flexible Approach to Automated RNN Architecture Generation (https://openreview.net/forum?id=SkOb1Fl0Z)

---

### Decision · Program_Chairs · 2018-01-29
**ICLR 2018 Conference Acceptance Decision**

**Decision:**

Invite to Workshop Track

**Comment:**

First off, this was a difficult paper to decide on. There was some vigorous discussion on the paper centering around the choices that were available to the conv-nets.  The author's strongly emphasized the improvements on the PTB task.

For my part, I think the method is very compelling -- sharing weights for all the models we are optimizing on seems like a great idea -- and that we can make it work is even more interesting. So from this point of view, I think its a novel contribution and worth accepting.

On the other hand, I'm likely to agree with some of the motivations behind the questions raised by R3. Are all the choices really necessary ? perhaps the gains came from just a couple of things like number of skip connections and channels, etc. That exploration is useful. On the flip side, I think it may be an irrelevant question -- the model is able to make the correct decisions from a big set.

The authors emphasize the language modelling part, but for me, this was actually less compelling. The authors use some of the tricks from Merity in their model training (perplexity 52.8), and as a result are already using some techniques that produces better results. Further, PTB is a regularization game -- and that's not really the claim of this paper. Although, one could argue that weight sharing between different models can produce an ensembling / regularization effect and those gains may show up on PTB. A much more compelling story would have been to show that this method works on a large dataset where the impact of the architecture cannot be conflated with controlling overfitting better.

As a result, this puts the paper on the fence for me; even though I very much like the idea. Polishing the paper and making a more convincing case for both the CNNs and RNNs will make this paper a solid contribution in the future.